# FEDDFQ : PERSONALIZED FEDERATED LEARNING BASED ON DATA FEATURE QUANTIFICATION

## ABSTRACT

Personalized federated learning is widely used for heterogeneous data distributions across clients. However, existing methods are difficult to measure and utilize these heterogeneities accurately. To address this issue, in this paper, we propose a novel and efficient method named **FedDFQ** which uses a customized Data Identity Extraction Module (DIEM) to dynamically generate metric proxies that quantify data heterogeneity across different local clients in a privacy-friendly manner. The metric proxies are used to assess the contributions of global parameter aggregation and personalized gradient backpropagation for each local client. In addition, we design a plug-and-play Automatic Gradient Accumulation Module (AGAM) that regularizes personalized classification layers with re-balanced gradients. We provide theoretical explanations and experimental results that validate the effectiveness of the proposed FedDFQ. With comprehensive comparisons to existing state-of-the-art approaches, FedDFQ outperforms them on two benchmark datasets in different heterogeneous scenarios. The code can be accessed at :[URL].

## 1 INTRODUCTION

Federated Learning (FL) is a distributed machine learning paradigm that allows multiple participants to cooperatively train models under the guidance of a central coordinator without sharing local data. This approach is widely used in healthcare Hao et al. (2020); Zhang et al. (2021) and the financial field Zheng et al. (2021); Liu et al. (2021). The traditional federated learning method such as FedAvg McMahan et al. (2017) obtains a global model by averaging the parameters uploaded from all clients, which is prone to be influenced by data heterogeneity and leads to significant performance degradation. Despite new frameworks have been introduced that ensure consistency between local and global optimization goals to obtain a more stable global model Li et al. (2019); Oh et al. (2021). However, these attempts fail to escape the paradigm of aggregating local parameters and average them to obtain global parameters, struggling with data heterogeneity, and the difference in data distribution between different clients may lead to suboptimal model aggregation with performance degradation (Yang et al., 2019).

Correspondingly, several personalized federated learning approaches have been put forward, which aim to improve the adaptability of local clients for their corresponding preserved tasks Arivazhagan et al. (2019); Fallah et al. (2020). For instance, FedRep Collins et al. (2021) separates the local clients model into shared structures and personalized structures so that the shared part can represent the global knowledge and the personal part is adapted to local data characteristics. However, existing approaches to personalized federated learning suffer from several shortcomings: most only focus on isolating the bias introduced by the heterogeneous distribution of other clients, rather than exploiting complementary information in heterogeneous data. Others attempt to exploit complementary information in heterogeneous data based on features from backbones with learnable parameters, which suffer from biases at the initial and training stages. It impairs the generalization to the local client on out-of-distribution samples.

To address these shortcomings, we propose a novel method, named FedDFQ, which quantifies the degree of data divergence among each client and uses it to measure the contribution of individual clients in this round. Specifically, we design a data representation extraction module to map local data into high-dimension semantic space for security and privacy. Then metric proxies are constructed through semantic relationships between mapped data to quantify heterogeneity across local clients.

They are used to evaluate the contributions of global parameter aggregation and personalized gradient backpropagation for each local client. In addition, we provide an efficient AGAM that regularizes personalized classification layers with re-balanced gradients. Finally, FedDFQ achieves state-of-the-art performance on two benchmark datasets and demonstrates significant potential with the increasing of clients. Our contributions are summarized as follows:

- We propose a novel method for extracting the heterogeneous representations of data distribution without introducing data biases. The representations are assembled to generate proxies that reflect the correlation of predictions of client models. We also prove that the metric proxies can fit the distribution of predictions by the proposed method.

- We develop a novel gradient integration module that regularizes the classification layers with gradients containing semantic relationships of heterogeneous data distributions, improving the generalization of clients with out-of-distribution samples. The method is play-and-plug and can be integrated into any existing federated learning paradigm.

- We conduct detailed experiments on two benchmark classification datasets. The proposed FedDFQ outperforms other state-of-the-art methods with different heterogeneous scenarios by large margins. Extensive results also demonstrate significant potential of the proposed method with the increase in clients.

The rest of this paper is organized as follows. Section 2 reviews the related works and the architecture of the proposed FedDFQ is described in Section 3. In Section 4, We present experimental results compared to other state-of-the-art methods and ablation studies to validate the effectiveness of our algorithm. Section 5 concludes this paper with remarks.

## 2 RELATED WORK

### 2.1 FEDERAL LEARNING WITH HETEROGENEOUS DATA

Federated learning aims to solve the problems of data privacy protection and data security, allowing multiple participants to jointly train machine learning models without sharing raw data. Traditional federated learning methods represented by FedAvg, construct global models by training the model locally at the client and sending the updated model parameters to a central server for average aggregation. However, these methods assume that the data distributions of all clients are independently identically distributed (IID), leading to poor performance of the model on certain clients in the presence of data heterogeneityYe et al. (2023). Therefore, Subsequent studies aim to optimize the aggregation process of model parameters by quantifying the feature similarity of client datasets. For example, FedProto proposed a prototype learning approach that allows clients to make personalized adjustments while maintaining global consistency Tan et al. (2022b). FedBABU enhances the model's adaptability under non-independently identically distributed (non-IID) data through an adaptive weight assignment mechanism Oh et al. (2021). Besides, FedProx Li et al. (2020) introduces the proximal term to restrict the updates of the local model that ensure similarity with the global model. FedDyn Acar et al. (2021) regularizes the local client model to reduce parameter drift with the global model and bring the local optima closer to the global optima. These methods not only improve the model's ability to adapt to local data features but also enhance the model's generalization performance in heterogeneous environments. However, the best performance for each local client may not be an optimal solution for the global model.

### 2.2 PERSONALIZED FEDERATED LEARNING

To overcome this limitation, the concept of personalized federated learning has been proposed, aiming to improve the local adaptation and overall performance of the model by adapting to the data distribution specific to each clientTan et al. (2022a). Ditto Li et al. (2021a) achieves better personalized performance by introducing a regularization term, which enables the client model to be locally fine-tuned while maintaining consistency with the global model. FedProto Tan et al. (2022b) introduces prototypical regularization to constrain the local updates, mitigating the effect of bias due to data distribution heterogeneity. FedPAC Xu et al. (2023b) leverages global semantic knowledge to align local-global features for better representations and quantifies the benefit of classifier combination for each client as a function of the combining weights, deriving an optimization

problem for estimating optimal weights. FedRep Collins et al. (2021) separates the local clients model into shared structures and personalized structures so that the shared part can represent the global knowledge and the personal part is adapted to local data characteristics. Recently, FedHKD Chen et al. (2023) applies prototype learning and knowledge distillation to train local clients for averaged representations and hyper-knowledge, instead of uploading model parameters. LG-Mix Jiang et al. (2024) utilizes the coverage rate of Neural Tangent Kernel to quantify the importance of local and global updates and mix these importance indices to update parameters. However, these methods neglect the utilization of interactions among clients with similar distributions to collaboratively train personalized classifiers to reduce the variance of client models.

## 3 METHOD ARCHITECTURE

We propose an efficient method to improve federated learning in heterogeneous data distribution scenarios. The overview of FedDFQ is shown in Figure 1. Specifically, we follow the design of one server to multi-clients. The input image first passes the feature extraction layer and data identification extraction in local clients, which aims to obtain abstract representations for classification and high-dimension distribution vectors. Then we pack up the global parameters, personalized gradients, and distribution vectors uploaded to the server. The distribution vectors generate metric proxies that assess contributions for each client in this aggregation. Next, the server distributes the global parameters re-balanced based on contributions in metric proxies. Finally, to further improve the generalization of clients to out-of-distribution samples, we design an AGAM that regularizes the classification layers with semantic-aware gradients for each client.

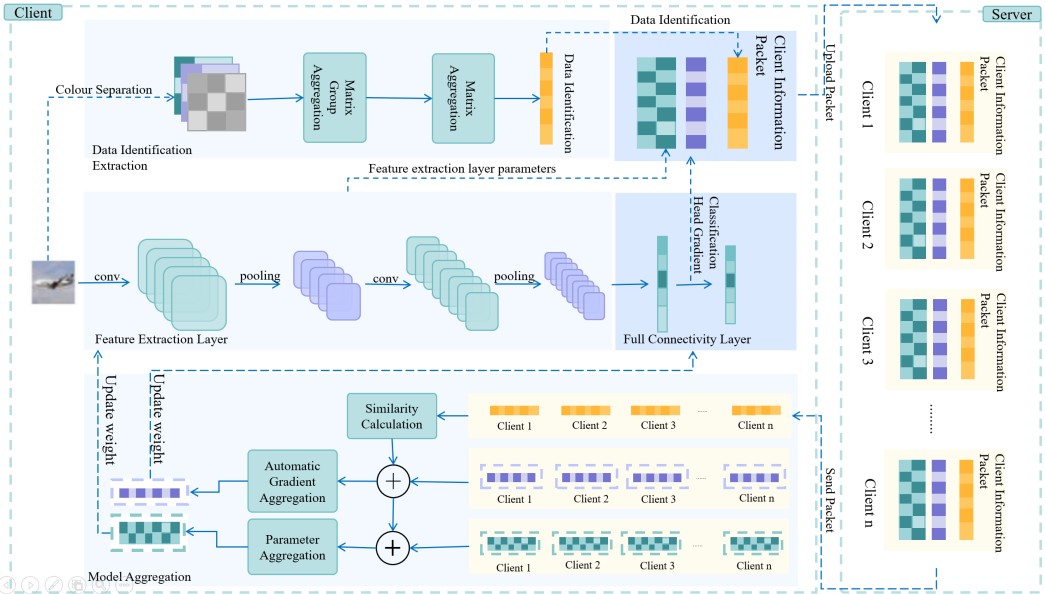

Figure 1: The architecture of FedDFQ.

### 3.1 DATA IDENTITY EXTRACTION MODULE

To evaluate the data heterogeneity of different local clients and follow the objectives of federated learning to protect the privacy and security of data, we propose the DIEM. Different from existing methods that use abstract representations from local feature extractors to denote the data distribution, these extractors with learnable parameters inevitably introduce undesirable data biases during the model initialization and training stage, leading to performance degradation with out-of-distribution samples.

The proposed method applies a no-parameter algorithm consisting of two data aggregation operations at local clients. Specifically, we first aggregate and mean input data along the channel dimension to obtain the 2D feature identifiers. Next, the second aggregation averages these 2D feature identifiers

to generate the final data representation vectors. This approach not only captures the basic attributes of the image data but also effectively represents the data distribution without introducing undesirable biases from learnable parameters.

We assume that the input image $x$ with shape $H \times W \times C$ dimensions. Where $H$ denotes the height, $W$ denotes the width, and $C$ denotes the channels of the image, respectively. The first data aggregations can be formulated as:

$$\mathbf{Y} = \frac{1}{C} \sum_{k=1}^{C} \mathbf{X_k}, \tag{1}$$

where $\mathbf{X}_k \in \mathbb{R}^{H \times W}$ denote the single channel image. Next, we compute the vector $\bar{\mathbf{x}}(j)$ for each column $j$ of the matrix $\mathbf{Y}$ as follows equation 2:

$$\bar{\mathbf{x}}_j = \frac{1}{H} \sum_{i=1}^{H} \mathbf{y}_i. \tag{2}$$

### 3.2 METRIC PROXIES

In the previous subsection, we detail the process of acquiring clients' data identifiers. To better understand the impact of data representations on local model aggregation, we aim to generate metric proxies that adaptively assess the contributions of each local model from the perspective of data distribution similarity. The most relative to data distribution is the predictions of the local client, which can be used to measure data heterogeneities with other local clients. Therefore, the heterogeneity can be formulated as follows:

$$S(\mathbf{z}_i, \mathbf{z}_j) = \text{Similarity}(\mathbf{z}_i, \mathbf{z}_j) = \frac{\mathbf{z}_i \cdot \mathbf{z}_j}{\|\mathbf{z}_i\| \|\mathbf{z}_j\|}, \tag{3}$$

where $\mathbf{z} \in \mathbb{R}^C$ denotes the class scores from classification layers with $C$ categories. We chose cosine similarity as the metric to reflect and quantify the heterogeneity with other clients. However, directly uploading predictions of local models conflicts with the objectives of federated learning. As an alternative, we pack up the data identifiers aggregated by the DIEM and upload them to the server. These data identifiers compose the metric proxies that indicate the quantified data heterogeneity of different clients. The formula is:

$$S(\bar{\mathbf{x}}_i, \bar{\mathbf{x}}_j) = \text{Similarity}(\bar{\mathbf{x}}_i, \bar{\mathbf{x}}_j) = \frac{\bar{\mathbf{x}}_i \cdot \bar{\mathbf{x}}_j}{\|\bar{\mathbf{x}}_i\| \|\bar{\mathbf{x}}_j\|}, \tag{4}$$

where the $\mathbf{z} \in \mathbb{R}^W$ is the data identity. However, why results of the DIEM can be an alternative to class scores? We present theoretical explanations to prove our algorithm. Because the similarity matrices of class scores and data identifiers are consistent with dimensions $N \times C \times C$, where the $N$ and $C$ represent the number of clients and the number of categories, in the semantic space, there is a matrix $\mathbf{A}$ with shape $N \times C \times C$ that the product of the matrix of data identifiers is equal to the matrix of class scores according to the theory of matrix. The formula is:

$$S(\bar{\mathbf{z}}_i, \bar{\mathbf{z}}_j) = \alpha S(\bar{\mathbf{x}}_i, \bar{\mathbf{x}}_j), \tag{5}$$

where $\alpha$ is the coefficient in the position $(i, j)$ of matrix $\mathbf{A}$. Equally, the class score $\mathbf{z}$ is represented as the product of data identifiers $\mathbf{x}$ and a weight metric $\mathbf{W}$ with shape $W \times C$ and addition of biases $b$. The formula is as follows:

$$\mathbf{z} = \mathbf{W}^T \mathbf{x} + \mathbf{b}. \tag{6}$$

Therefore, the conversion between two similarity matrices can be represented by a formula:

$$S(\mathbf{z}_i, \mathbf{z}_j) = \frac{\mathbf{x}_i^T \mathbf{W} \mathbf{W}^T \mathbf{x}_j + \mathbf{b}^T \mathbf{W}^T \mathbf{x}_i + \mathbf{b}^T \mathbf{W}^T \mathbf{x}_j}{\sqrt{\mathbf{x}_i^T \mathbf{W} \mathbf{W}^T \mathbf{x}_i + 2\mathbf{b}^T \mathbf{W}^T \mathbf{x}_i + \mathbf{b}^T \mathbf{b}} \cdot \sqrt{\mathbf{x}_j^T \mathbf{W} \mathbf{W}^T \mathbf{x}_j + 2\mathbf{b}^T \mathbf{W}^T \mathbf{x}_j + \mathbf{b}^T \mathbf{b}}} \tag{7}$$

We theoretically prove that the similarity of data identifiers can represent the data heterogeneity of different clients by linear projection. The specific reasoning process is detailed in the appendix.

## 3.3 PARAMETER AND GRADIENT INTEGRATION

To enhance the generalization of local models with out-of-distribution samples by leveraging heterogeneous data across clients, we propose aggregation modules that incorporate both global parameters aggregation and personalized gradient accumulation perspectives according to metric proxies.

### 3.3.1 GLOBAL PARAMETERS AGGREGATION

We aim to optimize the aggregation process by considering the heterogeneity of data across clients. Instead of simply averaging the parameters, as done in traditional methods like FedAvg, which disregards the data heterogeneity and results in significant performance degradation with out-of-distribution samples. Depending on the metric proxies that quantify heterogeneity with cosine similarity, the server aggregates the global parameters of clients. This approach allows FedDFQ to effectively utilize the heterogeneity of data across clients and mitigate the performance degradation typically observed in traditional aggregation methods. Here is the introduction to the algorithm process:

---

**Algorithm 1:** Feature Extraction Layer Parameter Aggregation (FELPA)

---

**Input** : The number of communication round $T$, the number of clients $N$, the distribution representation $\mathbf{F} = \{\mathbf{f}_0^t, \mathbf{f}_1^t, \ldots, \mathbf{f}_{N-1}^t\}$ and global parameters $\mathbf{W}^t = \{\mathbf{W}_0^t, \mathbf{W}_1^t, \ldots, \mathbf{W}_{N-1}^t\}$ in the round $t$

1 **begin**
2    **for** $t = 0, 1, \ldots, T-1$ **do**
3      **for** $i = 0, 1, \ldots, N-1$ **do**
4        Generate the metric proxy:
5        # Preserved for client $i$
6        $\mathbf{s}_i^t = \{s_{0,i}^t, s_{1,i}^t, \ldots, s_{N-1,i}^t\} = \{\mathbf{f}_i^t \times \mathbf{f}_0^t, \mathbf{f}_i^t \times \mathbf{f}_1^t, \ldots, \mathbf{f}_{N-1}^t \times \mathbf{f}_0^t,\}$
7        Normalize the similarities $\{s_{0,i}^t, s_{1,i}^t, \ldots, s_{N-1,i}^t\}$
8        Aggregate global parameters for clients $i$ based on the metric proxy:
9        $\mathbf{W}_i^t = \sum_{j=1}^N s_j^t \cdot \mathbf{W}_j^t$
10       Update parameters of local client $i$:
11       $\mathbf{W}_i^{t-1} \longleftarrow \mathbf{W}_i^t$
12      Collect re-balanced weight for $N$ clients:
13      $\mathbf{W}_{agg}^t = Aggregation(\mathbf{W}_0^t, \mathbf{W}_1^t, \ldots, \mathbf{W}_{N-1}^t)$

**Output** Aggregated weight $\mathbf{W}_{agg}^t$
:

---

### 3.3.2 PERSONALIZED GRADIENTS ACCUMULATION

To further improve the generalization of the proposed FedDFQ on out-of-distribution samples, we design an AGAM that regularizes personalized classification layers with re-balanced gradients. Specifically, the server utilizes metric proxies to re-scale the contribution of personalized gradients from other clients and then distributes all weighted gradients to their corresponding clients. It is worth noting that each client has its metric proxy, therefore, their gradient matrices are different, which ensures the personalization of local clients and leverages the heterogeneous gradient from other clients to improve the model's generalization. For local clients, they sequentially update the gradients of the personalized layer based on the similarity between clients, using the loss function generated from the train data as a criterion to measure the model performance. We finally select the gradients with mini-

mal loss values to update the parameters of local models. Here is the detail of the algorithm process:

---

**Algorithm 2:** Automatic Gradient Accumulation

---

**Input** : The number of communication round $T$, the number of clients $N$, the number of client gradients used in client $i$ $K_i$, the distribution representation $\mathbf{F} = \{\mathbf{f}_0^t, \mathbf{f}_1^t, \ldots, \mathbf{f}_{N-1}^t\}$ and personalized gradients $\mathbf{G}^t = \{\mathbf{G}_0^t, \mathbf{G}_1^t, \ldots, \mathbf{G}_{N-1}^t\}$ in the round $t$

1 **begin**
2    **for** $t = 0, 1, \ldots, T-1$ **do**
3      Server calculates metric proxies for each client:
4      **for** $i = 0, 1, \ldots, N-1$ **do**
5        $\mathbf{s}_i^t = \{s_{0,i}^t, s_{1,i}^t, \ldots, s_{N-1,i}^t\} = \{\mathbf{f}_i^t \times \mathbf{f}_0^t, \mathbf{f}_i^t \times \mathbf{f}_1^t, \ldots, \mathbf{f}_{N-1}^t \times \mathbf{f}_0^t, \}$
6        Normalize the similarities and sort in ascending order. $\{s_{0,i}^t, s_{1,i}^t, \ldots, s_{N-1,i}^t\}$
7        Re-scale gradients for each client:
8        $\mathbf{G}_{new,i}^t = \{\mathbf{G}_0^t \times s_{0,i}^t, \mathbf{G}_1^t \times s_{1,i}^t, \ldots, \mathbf{G}_{N-1}^t \times s_{N-1,i}^t\}$
9        Distribute gradients to corresponding clients:
10        client$_i \longleftarrow \mathbf{G}_{new,i}^t$
11    **for** each client $i \in [N]$ **in parallel do**
12      **for** $j = 1, 2, \ldots, k_i - 1$ **do**
13        try: $\mathbf{G}_{i,j-1}^t \longleftarrow \mathbf{G}_{i,j-1}^t + \mathbf{G}_j^t$
14        Compute loss on train set:
15        $L_{temp} = ClassificationLoss(\text{Pred, GT})$
16        **if** $L_{temp} < L_{min}$ **then**
17          $L_{min} = L_{temp}$
18          $\mathbf{G}_{i,j-1}^t \longleftarrow \mathbf{G}_{i,j}^t$
19        **else**
20          $\mathbf{G}_{i,j}^t \longleftarrow \mathbf{G}_{i,j-1}^t$

**Output** Updated model parameter $\mathbf{W}_i$ for each client
:

---

# 4 EXPERIMENTS

## 4.1 EXPERIMENTAL SETUP

**Datasets.** We use two widely used bechmark datasets CIFAR-10 Krizhevsky & Hinton (2009) and Fashion MNIST Xiao et al. (2017). Specifically, CIFAR-10 contains 60,000 color images of 32x32 size with 10 categories divided into 50,000 for training and 10,000 for testing. FashionMNIST consists of 60,000 training images and 10,000 testing images of 28x28 resolution with 10 categories. We utilize the Dirichlet distribution to model a non-independent identically distributed (non-IID) data partitioning strategy Yurochkin et al. (2019). Sample indices in the dataset are iteratively assigned to each client to ensure that each client has a minimum amount of data. By modeling the distribution of each client's data using the Dirichlet distribution, we can concentrate the data in specific categories, resulting in a non-uniform data distribution.

## 4.2 EXPERIMENTAL RESULTS

We compare our FedDFQ with other state-of-the-art methods with 50 and 100 clients on CIFAR-10 and FashionMNIST datasets. The results are shown in Table 1, we can observe that the traditional FL method FedAvg is suppressed by the local training by a large margin, which suffers from the heterogeneous data distribution. The FedDFQ outperforms the local training method, demonstrating that FedDFQ not only addresses the issue of data heterogeneity but also improves the robustness and generalization of local client models. In addition, compared with other FL approaches including new-optimization-based and personalization FL, FedDFQ achieves state-of-the-art performance on two benchmark classification datasets with 50 and 100 clients.

Table 1: The Performance Of FedDFQ And Other Methods On Two Benchmark Datasets.

| Method | CIFAR-10 | | FashionMNIST | |
|---|---|---|---|---|
| | 50 clients | 100 clients | 50 clients | 100 clients |
| LocalOnly | 87.97 | 86.63 | 97.34 | 94.92 |
| FedAvg McMahan et al. (2017) | 54.58 | 54.41 | 80.36 | 75.81 |
| SCAFFOLD Karimireddy et al. (2020) | 55.53 | 39.24 | 79.59 | 78.91 |
| Ditto Li et al. (2021a) | 87.89 | 86.28 | 96.96 | 94.57 |
| FedAMP Huang et al. (2021) | 87.84 | 86.54 | 97.32 | 95.76 |
| FedBN Li et al. (2021b) | 56.24 | 54.31 | 80.41 | 83.17 |
| FedFomo Zhang et al. (2020) | 87.45 | 86.02 | 97.30 | 95.57 |
| FedRep Collins et al. (2021) | 89.99 | 88.32 | 97.29 | 95.25 |
| FedBABU Oh et al. (2021) | 90.49 | 88.43 | 97.22 | 95.49 |
| FedProto Tan et al. (2022b) | 88.49 | 87.08 | 97.51 | 96.03 |
| FedCP Zhang et al. (2023b) | 90.06 | 89.06 | 97.50 | 95.81 |
| FedDBE Zhang et al. (2024) | 79.15 | 72.67 | 83.89 | 86.10 |
| FedGH Yi et al. (2023) | 87.83 | 86.58 | 97.33 | 95.87 |
| FedPAC Xu et al. (2023a) | 81.79 | 60.90 | 91.77 | 94.70 |
| GPFL Zhang et al. (2023a) | 85.21 | 86.43 | 97.26 | 95.84 |
| **FedDFQ** | **90.68** | **89.80** | **98.15** | **96.94** |

## 4.3 ABLATION STUDY

The FedDFQ algorithm consists of the DIEM, FELPA, and AGAM. We conduct ablation comparison experiments to validate the effectiveness of these components with 50 clients on the CIFAR-10 dataset and FashionMNIST datasets. The results are illustrated in Tabel 2, in which the **Local** means only local model training is performed; **FELPA** means only the FELPA is used; **AGAM** means only the AGAM is used; **w/o DIEM** means the average strategy is applied to aggregate parameters; **DIEM** means only the DIEM is used; **All** denotes three modules are all used.

Table 2: Ablation results of the proposed FedDFQ.

| Dataset | Local | FELPA | AGAM | w/o DIEM | DIEM | All |
|---|---|---|---|---|---|---|
| **CIFAR-10** | 88.07 | 85.18 | 88.09 | 72.66 | 75.79 | **90.46** |
| **FashionMNIST** | 97.32 | 90.42 | 97.46 | 90.12 | 90.93 | **98.15** |

The experimental results prove the effectiveness of the three components in FedDFQ. Specifically, we can observe that the local training only achieves 88.07% and 97.32% Acc on CIFAR-10 and Fashion-MNIST datasets. After integrating the **FELPA** and **AGAM**, FedDFQ outperforms other designs by a large margin, because the AGAM is prone to fall into the local optimal solution and FELPA causes semantic gaps that significantly hinder classification layers from adapting aggregated parameters from other heterogeneous clients. The feature extraction layer can aggregate the parameters of other clients to provide momentum support for model parameter updating and help the AGAM jump out of the local optimal solution. Then the DIEM is integrated into FedAvg (**DIEM**) and outperforms vanilla FL (**w/o DIEM**) by 3.13 % Acc and 0.71% Acc on CIFAR-10 and FashionMNIST datasets.

In addition, we visualize the relationship between communication rounds and accuracy by line graphs. As shown in Figure 2, the accuracy improvement curve of the design that incorporates all modules (represented by the green line) exhibits a smoother trend when compared to other methods, which proves that the components in FedDFQ keep the consistency optimization goals between local and global levels and lead to a more stable training process. In more detail, the green line in Figure 2a is significantly ahead compared to Figure 2b, indicating that our method outperforms other approaches, particularly in handling more complex data distributions.

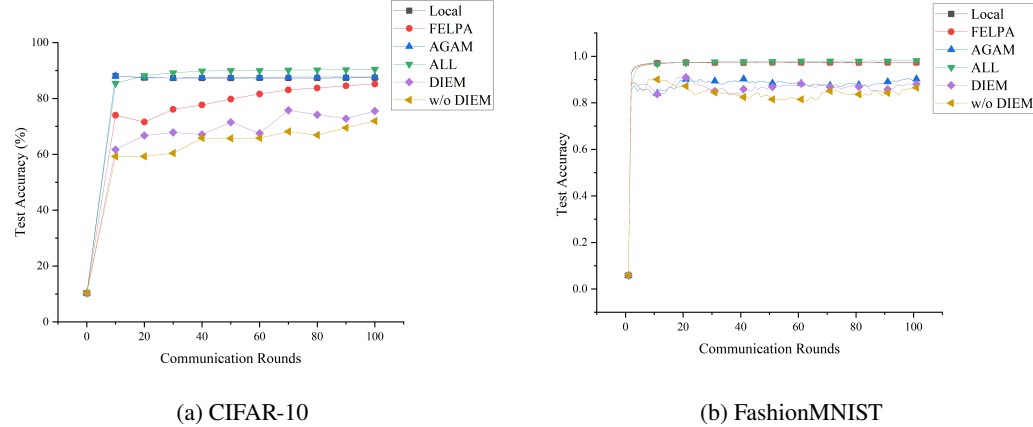

(a) CIFAR-10             (b) FashionMNIST

Figure 2: The Convergence curves of FedDFQ and other state-of-the-art methods on CIFAR-10 and FashionMNIST datasets.

### 4.4 ANALYSIS STUDY

We analyze the proposed FedDFQ on the CIFAR-10 dataset under different numbers of clients. As shown in Figure 3, our method surpasses the locally trained model when the number of clients is 50, 75, and 100. Furthermore, as the number of clients increases, our method demonstrates even more significant advantages. This demonstrates that our approach not only addresses data heterogeneity but also learns from similar data distributions in other clients to enhance the generalization capability of the local model.

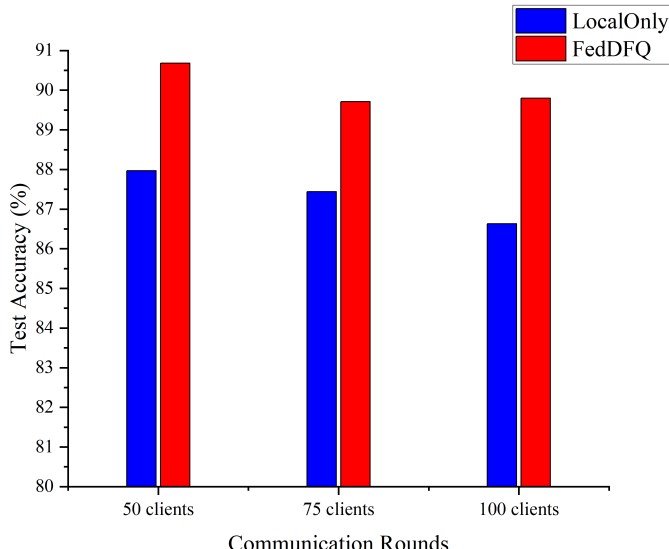

Figure 3: The performance of FedDFQ compared to local training strategy with different numbers of clients.

To validate the robustness of FedDFQ, we select three types of data distribution to construct heterogeneous datasets. The distribution is detailed as follows:

- **d1** denotes Dirichlet distribution to partition the dataset. It starts by determining the number of categories $K$ and selecting a parameter vector $\boldsymbol{\alpha}$ (such as $\alpha_i = 1$). Samples are drawn from

Dirichlet ($\alpha$) to obtain a probability vector $\boldsymbol{\theta}$, where the elements sum to 1, representing the proportion of each category.

- **d2** is pathological distribution, which is a Non-IID, and unbalanced data set partitioning mode. There are 50 clients in the test, and the data set of each client covers two categories, but the sample quantity of these two categories is greatly different. Data set partitioning is achieved by creating an index array according to category allocation rules. Each client allocates a fixed number of categories, and category allocation follows a certain logical order.

- **d3** is a Non-IID data set partitioning mode, with a total of 50 clients, and each client's data set covers all 10 categories, but the number of samples in different categories varies significantly between clients. Data set partitioning is also achieved by creating indexed arrays and assigning them according to class allocation rules, ensuring that each client covers all classes, but the sample number for each class is unevenly distributed.

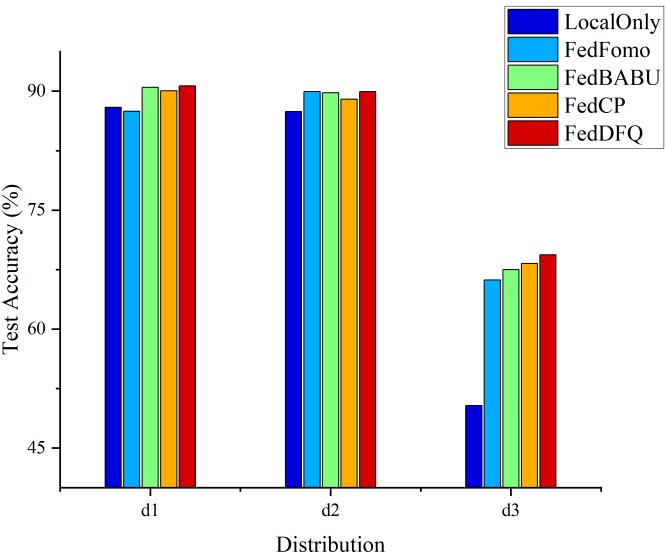

Figure 4: Performance of FedDFQ under different data distributions.

As illustrated in Figure 4, we can observe that although our method exhibits fluctuations in performance under different distributions, it consistently outperforms other methods. This demonstrates that FedDFQ is capable of adapting to heterogeneous data with different distributions, and the local model possesses stronger robustness and generalization capability.

## 5 Conclusion and future work

In this paper, we propose a novel personalized federated learning method named FedDFQ including IDEM, FELPA, and AGAM modules. Specifically, the IDEM measures and quantifies data heterogeneity for each client to generate metric proxies, which re-scale the contributions of local clients in the current rounds. FELPA aggregates weighted parameters of feature extractors according to metric proxies from each client. To further utilize the homogeneity in heterogeneous clients, AGAM is designed to regularize personalized classification layers with weight gradients, improving the generalization of local models. Besides, we conduct extensive experiments and provide theoretical proof to validate the effectiveness of our FedDFQ, and our method achieves state-of-the-art performance on two benchmark datasets.

Future work includes developing more refined methods for extracting distributed features and conducting experimental analyses on more datasets.

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

# A APPENDIX

## A.1 METRIC PROXIES THEORY PROOF

$$S(\mathbf{z}_i, \mathbf{z}_j) = \frac{\mathbf{x}_i^T \mathbf{W} \mathbf{W}^T \mathbf{x}_j + \mathbf{b}^T \mathbf{W}^T \mathbf{x}_i + \mathbf{b}^T \mathbf{W}^T \mathbf{x}_j}{\sqrt{\mathbf{x}_i^T \mathbf{W} \mathbf{W}^T \mathbf{x}_i + 2\mathbf{b}^T \mathbf{W}^T \mathbf{x}_i + \mathbf{b}^T \mathbf{b}} \cdot \sqrt{\mathbf{x}_j^T \mathbf{W} \mathbf{W}^T \mathbf{x}_j + 2\mathbf{b}^T \mathbf{W}^T \mathbf{x}_j + \mathbf{b}^T \mathbf{b}}} \quad (8)$$

When $\mathbf{W}$ is the identity matrix and $\mathbf{b}$ is the zero vector,

$$S(\mathbf{z}_i, \mathbf{z}_j) = \frac{\mathbf{x}_i^T \mathbf{x}_j}{\sqrt{\mathbf{x}_i^T \mathbf{x}_i} \cdot \sqrt{\mathbf{x}_j^T \mathbf{x}_j}} = S(\mathbf{x}_i, \mathbf{x}_j) \quad (9)$$

To further verify the correlation between $S(\mathbf{z}_i, \mathbf{z}_j)$ and $S(\mathbf{x}_i, \mathbf{x}_j)$, we collect $S(\mathbf{z}_i, \mathbf{z}_j)$ and $S(\mathbf{x}_i, \mathbf{x}_j)$ data through experiments and calculate their Pearson correlation coefficient to demonstrate their significant correlation.

Table 3: Statistics Collected On CIFAR-10 Will Be
Open Sourced Along With The Source Code

|  | Average Value | Variance |
|---|---|---|
| $S(\mathbf{z}_i, \mathbf{z}_j)$ | 0.999619387 | 0.000000171 |
| $S(\mathbf{x}_i, \mathbf{x}_j)$ | 0.473363243 | 0.600071219 |

Plug the resulting statistics into equation 10.

$$r = \frac{\frac{1}{n} \sum_{i=1}^{n} (X_i - \bar{X})(Y_i - \bar{Y})}{\sqrt{\frac{1}{n} \sum_{i=1}^{n} (X_i - \bar{X})^2} \cdot \sqrt{\frac{1}{n} \sum_{i=1}^{n} (Y_i - \bar{Y})^2}} = 0.728315558 \quad (10)$$

Through the experimental data, we can see that there is a strong correlation between $S(\mathbf{z}_i, \mathbf{z}_j)$ and $S(\mathbf{x}_i, \mathbf{x}_j)$

## A.2 EXPERIMENT DETAILS SETUP AND RESULT VISUALIZATION

### A.2.1 MODELS

In the experiment, we used a 3-layer convolutional neural network(CifarCNN) for the CIFAR-10 dataset and a 2-layer convolutional neural network(MNISTCNN) for the FashionMNIST dataset. The specific network structure is as follows.

```
CifarCNN(
  (conv1): Conv2d(3, 16, kernel_size=(5, 5), stride=(1, 1))
  (relu1): ReLU()
  (pool): MaxPool2d(kernel_size=2, stride=2, padding=0,
                    dilation=1, ceil_mode=False)
  (conv2): Conv2d(16, 32, kernel_size=(5, 5),
                    stride=(1, 1), padding=(1, 1))
  (relu2): ReLU()
  (conv3): Conv2d(32, 64, kernel_size=(3, 3),
                    stride=(1, 1), padding=(1, 1))
  (relu3): ReLU()
  (fc1): Linear(in_features=576, out_features=128, bias=True)
  (fc): Linear(in_features=128, out_features=10, bias=True)
)

MNISTCNN(
  (conv1): Sequential(
```

```
    (0): Conv2d(1, 32, kernel_size=(5, 5), stride=(1, 1))
    (1): ReLU(inplace=True)
    (2): MaxPool2d(kernel_size=(2, 2), stride=(2, 2),
                   padding=0, dilation=1, ceil_mode=False)
  )
  (conv2): Sequential(
    (0): Conv2d(32, 64, kernel_size=(5, 5), stride=(1, 1))
    (1): ReLU(inplace=True)
    (2): MaxPool2d(kernel_size=(2, 2), stride=(2, 2),
                   padding=0, dilation=1, ceil_mode=False)
  )
  (fc1): Sequential(
    (0): Linear(in_features=1024, out_features=512, bias=True)
    (1): ReLU(inplace=True)
  )
  (fc): Linear(in_features=512, out_features=10, bias=True)
)
```

### A.2.2 EXPERIMENTAL DETAILED SETTINGS

Our experiments are conducted on the NVIDIA GeForce RTX 3060Ti GPUs. We set the number of local training rounds to 5 for the clients and the number of global communication rounds to 100. All the clients participate in the training. The batch size is kept at 64 and the learning rate is 0.01 with SGD optimizer Ruder (2016) during local training stages.

### A.2.3 ADDITIONAL EXPERIMENTAL RESULTS

We use line plots to show the specific performance of different methods in the CIFAR-10 dataset. As shown in Figure 6a. It is worth mentioning that we put all the clients into training, the purpose of which is to see how FedDFQ performance changes in a multi-client situation. We can observe that our method exhibits a smoother trend when compared to other methods with 50 and 100 clients, which proves that the components in FedDFQ keep the consistency optimization goals between local and global levels and lead to a more stable training process. In addition, under the same number of iterations, our method achieves a higher accuracy, demonstrating its superiority in terms of convergence.

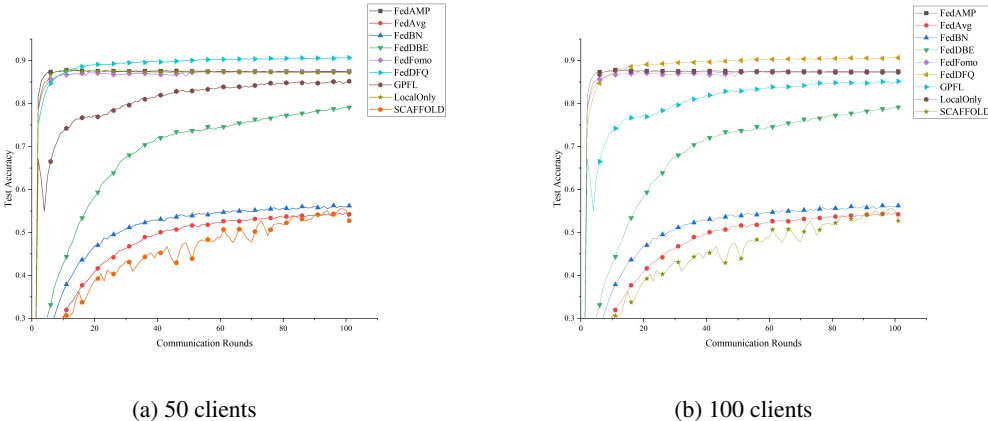

(a) 50 clients            (b) 100 clients

Figure 5: Test Accuracy Under Different Training Methods And Different Clients

In our experiments, we compare homogeneous and heterogeneous distributions on the FashionMNIST dataset. As shown in Figure 6. Our FedDFQ outperforms other methods both in iid. and no-iid conditions. Furthermore, the final accuracy of FedDFQ on no-iid data suppresses other designs With a significant advantage, demonstrating our method can learn generalized information from gathered parameters and gradients, and lead to the superiority of adaptability to heterogeneous data.

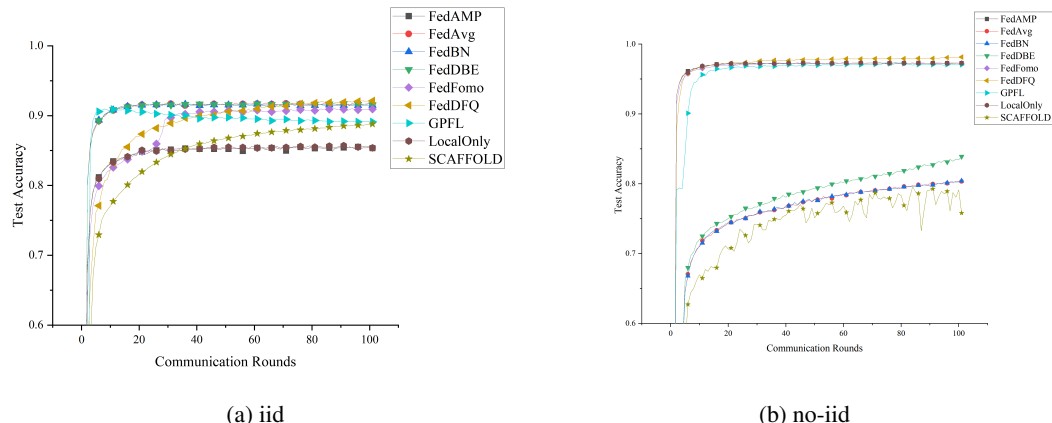

(a) iid

(b) no-iid

Figure 6: Test accuracy with different training methods and different data distributions

