# OpenReview forum: "FedDFQ : Personalized Federated Learning Based On Data Feature Quantification"
_ICLR.cc/2025/Conference — ICLR 2025 Conference Withdrawn Submission_

### Official Review · Reviewer_Bqvr · 2024-10-28

**Soundness:** 2
**Presentation:** 3
**Contribution:** 2
**Rating:** 3
**Confidence:** 4

**Summary:**

This paper introduces a novel method of personalized federated learning named FedDFQ, which is specifically designed to address the issue of data heterogeneity across client datasets. The method employs a Data Identity Extraction Module (DIEM) to dynamically generate metric proxies that quantify the degree of data divergence among different local clients. These metric proxies are then utilized to assess the contributions of global parameter aggregation and personalized gradient backpropagation for each local client. Furthermore, the paper presents an Automatic Gradient Accumulation Module (AGAM) that regularizes the personalized classification layers using re-balanced gradients.

**Strengths:**

1.The paper is well-organized and the language is clear.

2.The paper addresses a novel aspect of federated learning.

3.The article includes comprehensive experiments, including ablation studies, and the results are compelling.

**Weaknesses:**

1. The introduction of the method seems somewhat abrupt, lacking sufficient context, making it difficult to understand the motivation behind the FedDFQ approach.

2.The experimental setup section is too brief, lacking details on baseline selection, evaluation metrics, and hardware specifications.

3.The comparison in the experimental results section is too general, lacking a discussion of the data between FedDFQ and each of the other distinct Federated Learning methods, and only one dataset is compared.

4.In Figure 1, the arrows are randomly directed and not aligned properly, and the text formatting within the figure is not easy to read.

**Questions:**

What is the rationale for selecting the experimental baselines? Why compare the performance of FedDFQ with baseline methods on the two benchmark classification datasets with 50 and 100 clients?

---

### Official Review · Reviewer_K1cL · 2024-10-30

**Soundness:** 2
**Presentation:** 1
**Contribution:** 1
**Rating:** 3
**Confidence:** 4

**Summary:**

This paper proposes a new personalized federated learning algorithm, dubbed FedDFQ. Aiming to debias the local personalization with information from other clients, FedDFQ uses a customized Data Identity Extraction Module (DIEM) which enables a personalized aggregation strategy for the model weights and gradients on each client. Experiments show the superiority of FedDFQ over compared algorithms.

**Strengths:**

- The problem that this paper focuses on is important.
- Compared methods in the experiments are plenty.
- Ablation study is included, which proves the effectiveness of different modules in the proposed algorithm.

**Weaknesses:**

**Major:**
- The presentation of the method needs to be improved. I personally find it hard to understand for a lot of sentences.
- The novelty is limited the so called "Data Identification Extraction Module (DIEM)" is just an averaging of a $H \times W \times C$ image over the $H$ and $C$ dimension into a $W$-dimensional vector.
- The communication overhead for the proposed federated learning algorithm is extremely high. Each client needs to upload a packet including the model weights and gradient and the data identification vector, and download the packets from all other clients.
- The motivation of the proposed DIEM is not clear. Why can't the distribution representation base on the feature of the global model? What bias would that introduce?
- I am confused with how the matrix $A$ with shape $N \times C \times C$ forms. There is no explanation on it. In general I have problem with the what Sec 3.2 is trying to say. Is $S(z_i, z_j)$ or $S(x_i, x_j)$ the similarity between two images or two images or two clients. Since $S(z_i, z_j)$ or $S(x_i, x_j)$ are two scalars, of course there is a coefficient $\alpha$ that makes $S(z_i, z_j)= \alpha S(x_i, x_j)$ holds. What is this equation (Eq. (5)) trying to say here? And how does matrix $A$ come in to play here? Is its shape $(N \times C) \times C$ or $N \times (C \times C)$?. Otherwise, I believe you should refer $A$ as a tensor instead of a matrix.
- Line 215 says you are "theoretically prove that the similarity of data identifiers can represent the data heterogeneity of different clients". There is no need to prove that unless you have a specific mathematical equations that you are trying to prove? Besides, in you proof, since you have Eq(6) already ($z=W^T x + b$), how can you simply let $W$ and $b$ be identity matrix and zero vector to say $S(z_i, z_j)=  S(x_i, x_j)$?

**Minor:**
- The criticism on existing personalized federated learning is not clear (line 45-49). There is little expiation and no citation the claimed criticism.
- No experiments on different sampling rate.
- Too many typos, even in the equations. Please proofread.

**Questions:**

Please see questions in the weaknesses.

---

### Official Review · Reviewer_mGur · 2024-10-31

**Soundness:** 1
**Presentation:** 1
**Contribution:** 1
**Rating:** 1
**Confidence:** 4

**Summary:**

This paper introduces a personalized federated learning approach named FedDFQ, aimed at addressing data heterogeneity issues across clients. FedDFQ comprises three main components: the Data Identity Extraction Module (DIEM), the Feature Extraction Layer Parameter Aggregation module (FELPA), and the Automatic Gradient Accumulation Module (AGAM). DIEM generates metric proxies by aggregating data channel features to quantify data heterogeneity among clients. FELPA performs weighted parameter aggregation based on client data characteristics, while AGAM regularizes the personalized classification layer with re-balanced gradients to improve the model's generalization to out-of-distribution samples. Experiments were conducted on two benchmark datasets, CIFAR-10 and FashionMNIST, demonstrating FedDFQ's superiority in different heterogeneous scenarios, especially as the number of clients increases.

**Strengths:**

The proposed Automatic Gradient Accumulation Module (AGAM) aims to regularize the classification layer through personalized gradient accumulation, enhancing the model's generalization capability on out-of-distribution samples. This module demonstrates the paper's effort to improve the model's generalization ability, which holds practical value. Additionally, AGAM has a plug-and-play feature, allowing for easy integration into existing federated learning frameworks.

**Weaknesses:**

This work introduces FedDFQ, yet the approach lacks significant innovation and makes limited contributions compared to existing federated learning methods. The main approach of using channel summation to extract client data quality and cosine similarity for data heterogeneity measurement is straightforward but does not substantially advance the field.

The authors state that “we first aggregate and mean input data along the channel dimension to obtain the 2D feature identifiers”. This approach to data extraction through simple channel summation is overly simplistic and results in significant information loss. This method does not capture nuanced features, which may lead to suboptimal heterogeneity quantification.

The metric proxies rely on “cosine similarity as the metric to reflect and quantify the heterogeneity with other clients”. The reliance on cosine similarity as the evaluation metric for data heterogeneity is overly simplistic, failing to capture complex variations across different client data distributions.

The paper briefly describes AGAM as “an efficient AGAM that regularizes personalized classification layers with re-balanced gradients” but does not provide any specific equation descriptions, leaving the AGAM’s implementation vague.

Metric proxies are generated from aggregated data and shared with the server. Unlike traditional federated learning methods, FedDFQ transmits additional distribution information between clients and the server, potentially impacting data privacy. However, the paper does not address or mitigate these privacy concerns no matter in theoretical analysis or methods.

While FELPA and AGAM are proposed for parameter aggregation and gradient accumulation, there is no theoretical support provided regarding the stability or convergence of the method under these modifications.

The experimental validation is weak, with tests conducted only on two relatively simple datasets (CIFAR-10 and FashionMNIST), which limits the generalizability and robustness of the method.

Table 2 in Section 4.3 shows that individual components perform worse than local training, while the combination improves results. It means that each component of FedDFQ is individually detrimental compared to local training, with only combined components producing positive results. The paper lacks an in-depth explanation for this phenomenon.

The paper’s formatting is rough, which negatively impacts readability and gives an unpolished impression.

**Questions:**

Can the authors provide privacy-preserving methods or guarantees for the extra distribution information transmitted?

Could the authors include convergence proofs for the weighted aggregation and regularization methods?

Would the authors consider expanding the experiments to larger-scale datasets for better validation?

Can the authors further explore how data heterogeneity impacts personalized federated learning and explain their current approach in more depth?

Could the authors expand the related work section to include more studies on heterogeneous federated learning?

---

### Note · Authors · 2024-11-14

**Comment:**

Thanks to the reviewers for their guidance and help, we will learn from this experience to improve this article!

**Withdrawal Confirmation:**

I have read and agree with the venue's withdrawal policy on behalf of myself and my co-authors.